# Evaluation of the Antioxidant Activity of Levetiracetam in a Temporal Lobe Epilepsy Model

**DOI:** 10.3390/biomedicines11030848

**Published:** 2023-03-10

**Authors:** Iván Ignacio-Mejía, Itzel Jatziri Contreras-García, Julieta Griselda Mendoza-Torreblanca, Omar Noel Medina-Campos, José Pedraza-Chaverri, Mercedes Edna García-Cruz, Antonio Romo-Mancillas, Saúl Gómez-Manzo, Cindy Bandala, María Elena Sánchez-Mendoza, Luz Adriana Pichardo-Macías, Noemí Cárdenas-Rodríguez

**Affiliations:** 1Laboratorio de Farmacología de Plantas Medicinales Mexicanas, Escuela Superior de Medicina, Instituto Politécnico Nacional, Mexico City 11340, Mexico; 2Laboratorio de Fisiología, Escuela Militar de Graduados de Sanidad, UDEFA, Mexico City 11200, Mexico; 3Laboratorio de Neurociencias, Instituto Nacional de Pediatría, Secretaría de Salud, Mexico City 04530, Mexico; 4Departamento de Biología, Facultad de Química, Universidad Nacional Autónoma de México, Mexico City 04510, Mexico; 5Laboratorio de Diseño Asistido por Computadora y Síntesis de Fármacos, Facultad de Química, Universidad Autónoma de Querétaro, Centro Universitario, Querétaro 76010, Mexico; 6Laboratorio de Bioquímica Genética, Instituto Nacional de Pediatría, Secretaría de Salud, Mexico City 04530, Mexico; 7Escuela Superior de Medicina, Instituto Politécnico Nacional, Mexico City 11340, Mexico; 8Departamento de Fisiología, Instituto Politécnico Nacional, Escuela Nacional de Ciencias Biológicas, Mexico City 07738, Mexico

**Keywords:** levetiracetam, temporal lobe epilepsy, antioxidant, neuroprotection

## Abstract

Epilepsy is a neurological disorder in which it has been shown that the presence of oxidative stress (OS) is implicated in epileptogenesis. The literature has shown that some antiseizure drugs (ASD) have neuroprotective properties. Levetiracetam (LEV) is a drug commonly used as an ASD, and in some studies, it has been found to possess antioxidant properties. Because the antioxidant effects of LEV have not been demonstrated in the chronic phase of epilepsy, the objective of this study was to evaluate, for the first time, the effects of LEV on the oxidant–antioxidant status in the hippocampus of rats with temporal lobe epilepsy (TLE). The in vitro scavenging capacity of LEV was evaluated. LEV administration in rats with TLE significantly increased superoxide dismutase (SOD) activity, increased catalase (CAT) activity, but did not change glutathione peroxidase (GPx) activity, and significantly decreased glutathione reductase (GR) activity in comparison with epileptic rats. LEV administration in rats with TLE significantly reduced hydrogen peroxide (H_2_O_2_) levels but did not change lipoperoxidation and carbonylated protein levels in comparison with epileptic rats. In addition, LEV showed in vitro scavenging activity against hydroxyl radical (HO•). LEV showed significant antioxidant effects in relation to restoring the redox balance in the hippocampus of rats with TLE. In vitro, LEV demonstrated direct antioxidant activity against HO•.

## 1. Introduction

Epilepsy is a brain condition that causes recurring, unprovoked seizures. The World Health Organization [1] defined epilepsy as a chronic noncommunicable disease that results from excessive electrical discharges in a group of brain cells [1]. According to the International League Against Epilepsy (ILAE), the definition of epilepsy requires any of the following conditions: (1) at least two unprovoked (or reflex) seizures occurring >24 h apart; (2) one unprovoked (or reflex) seizure and a probability of further seizures, similar to the general recurrence risk (at least 60%) after two unprovoked seizures, occurring over the next 10 years; and (3) diagnosis of an epilepsy syndrome [2]. Currently, epilepsy affects the lives of 10% of the global population; of these, 125,000 die each year, and over 80% of these deaths occur in low- and middle-income countries [3,4]. Oxidative stress (OS) is defined as an imbalance between oxidant species, such as reactive oxygen species (ROS) and/or reactive nitrogen species (RNS), and intra- and extracellular components of the antioxidant defense system, such as antioxidant enzymes and proteins or compounds with antioxidant activity, which disrupts redox signaling and control and/or causes molecular damage [5]. OS is involved in chronic diseases, including epilepsy, which are associated with neuronal hyperexcitation [6,7]. Oxidative/nitrosative stress and mitochondrial dysfunction are presented in all types of epilepsy, including absence epilepsy [8]. In a murine genetic model of absence epilepsy, it was shown that in cerebrospinal fluid and serum, the lipid peroxidation as oxidant marker was increased and superoxide dismutase (SOD) glutathione peroxidase (GPx) and total antioxidant capacity were decreased, indicating that redox balance can lead to a better diagnosis and treatment of this condition [9]. It is important to mention that emerging studies have shown that OS, ROS, or RNS and inflammation promote neuronal hyperexcitability and seizures and are crucial factors in the onset and progression of neurodegeneration in temporal lobe epilepsy (TLE), which is one of the most common epilepsies of focal origin [10,11,12,13,14]. The mechanisms underlying epileptogenesis also include the generation of ROS together with altered genes expression, inflammation, protein production, and changes in connectivity [15,16].

Antiseizure drugs (ASD) are used to treat epilepsy, and some of them, for example, levetiracetam (LEV) and valproic acid (VPA), have recently shown antioxidant properties related to their neuroprotective effect [17,18,19,20]. About VPA, our work group showed, in a longitudinal study, that VPA had antioxidant properties in epileptic children. Epilepsy in these children caused a decrease in the plasmatic activity of the antioxidant enzymes glutathione reductase (GR), superoxide dismutase (SOD), glutathione peroxidase (GPx), and catalase (CAT), and an increase in glutathione peroxidase (GPx), as well as an increase in the oxidant markers malondialdehyde (MDA), hydrogen peroxide (H_2_O_2_), 8-hydroxy-2-deoxyguanosine (8-OHdG), and 3 nitrotyrosine (3-NT) over one year of treatment. However, with the use of VPA as a monotherapy, GR, SOD, and CAT activities increased, GPx decreased, and all oxidant markers decreased [21]. With regard to the latter finding, ASDs could exhibit antioxidant activity, which would provide effective and neuroprotective results through ROS modulation.

LEV (*S*-enantiomer pyrrolidine derivative of α-ethyl-2-oxo-1-pyrrolidine acetamide IUPAC, (*S*)-2-(2- oxopyrrolidin- 1-yl) butanamide) is a second-generation ASD used for treatment of focal-onset, myoclonic, and generalized-onset seizures, refractory epilepsy, and for seizure cessation in patients with *status epilepticus* (SE). Its primary mechanism of action is modulation of SV2A (an integral membrane protein found in the vesicles of almost all synaptic terminals) [22,23,24]. LEV is well-tolerated and effective compared to other ASDs [25]. Furthermore, it has also shown anti-inflammatory, anti-ictogenic, and neuroprotective therapeutic properties [26]. Clinical and experimental studies have found that LEV can exert antioxidant properties [18,20,26,27,28,29,30,31].

In recent years, it has become clear that the current approach to treating epilepsy with ASDs that control seizures alone is insufficient because it does not address all the comorbidities presented by epileptic patients. Therefore, research to identify new ASDs or new properties of existing ASDs has been encouraged. LEV has been an effective drug in patients with TLE and has gradually become a drug of first choice because of its effectiveness and tolerability. Although there are some studies concerning the early antioxidant effect of LEV (in the SE or epileptogenesis period), its effect during the chronic phase of the disease has not been reported. Thus, the objective of this study was to evaluate, for the first time, the effects of LEV on the oxidant–antioxidant status in the hippocampus of rats with TLE by determining the profiles of the antioxidant activities (CAT, SOD, GPx, and GR) and oxidant markers levels (MDA; carbonylated proteins and H_2_O_2_). Additionally, we used spectrometric methods to evaluate the in vitro ferric reducing antioxidant power (FRAP), oxygen radical absorbance capacity (ORAC), hydroxyl radical (HO•), superoxide radical anion (O_2_^•−^), and 2,2-diphenyl-1-picrylhydrazyl radical (DPPH) scavenging capacity of LEV. 

## 2. Materials and Methods

### 2.1. Animals

Male Wistar rats (250–300 g; CINVESTAV, Mexico City, Mexico) were housed under standard conditions: controlled temperature (22 ± 2 °C) and a 12 h dark–light cycle (lights on at 6:00 a.m.), with food and water ad libitum. The rats were randomly allocated to one of four groups: control (CTRL), epileptic (EPI), control with levetiracetam (CTRL + LEV), and epileptic with levetiracetam (EPI + LEV). All procedures were performed in accordance with the NIH Guide for the Care and Use of Experimental Animals and Mexican law (SAGARPA NOM-062-ZOO-1999). All protocols were approved by the Local Institutional Committees, INP (registered number INP 2022/048).

### 2.2. Li^2+^-Pilocarpine-Induced Temporal Lobe Epilepsy (TLE) 

TLE was induced in the rats by the intraperitoneal (i.p.) administration of lithium chloride (127 mg/kg, Sigma, St. Louis, MO, USA) 19 h before pilocarpine administration. On the day of SE induction, the animals were injected with scopolamine methyl bromide (1 mg/kg i.p., Sigma, St. Louis, MO, USA) to minimize the peripheral cholinergic side effects and, 30 min later, they received a dose of pilocarpine hydrochloride (30 mg/kg i.p.; Sigma, St. Louis, MO, USA) [32,33]. If, after one hour, the animals did not show SE, a second dose of pilocarpine was administered; if they again failed to present SE, they were eliminated from the study. To determine the onset of SE, the animals were placed in acrylic cages, and their behavior was observed before and after the pilocarpine injection. SE was defined as continuous convulsive behavior for more than 5 min. The duration of the SE was 90 min. To terminate it, the rats received an intramuscular injection of diazepam 5 mg/kg and were placed on an ice bed for 1 h to mitigate the hyperthermia produced by SE. A second dose of diazepam (PISA, Mexico City, Mexico) was administered 8 h later, and the rats were then housed overnight in a room at 17 ± 2 °C. One day after SE induction, the room temperature was restored to 22 ± 2 °C. The post-SE care of the animals included feeding with a commercial dietary supplement twice a day for 3 days, and they were monitored constantly until their recovery [34].

### 2.3. Monitoring of Spontaneous Behavioral Seizures

Twelve weeks after SE induction, video monitoring of the animal behavior was carried out to ensure that the rats presented spontaneous recurrent seizures (SRS) and were therefore epileptic. For this, the animals were housed in individual acrylic cages and were continuously (24 h/7 days) video monitored using a 2-camera system (Steren Model CCTV-970, Mexico) for 2 weeks. Using the fast-forward speed (eight times the normal speed) of the video recorder, well-trained investigators who were blinded to the experimental groups manually reviewed the videos. We considered that animals exhibited an SRS when the Racine score reached ≥3 [33].

### 2.4. Levetiracetam Treatment

Osmotic pumps (Alzet, 2ML1, Cupertino, CA, USA; release rate 10 µL/h) were implanted in the CTRL + LEV and EPI + LEV groups at week thirteen. LEV was extracted from 1 tablet (1000 mg; Pharmalife Laboratories, Atlanta, GA, USA) and dissolved in 3 mL of 0.9% NaCl (PISA, Mexico City, Mexico). The solution was sonicated for 10 min, centrifuged, and filtered (0.45 µm; Corning, NY, USA) before use [35]. Then, the osmotic pumps were filled with 2 mL of LEV and incubated for 4 h at 37 °C in 0.9% NaCl. Next, the rats were anesthetized with isoflurane (Sofloran^®^Vet, PISA, Mexico City, Mexico) and the pumps were implanted subcutaneously. Finally, an acute dose of LEV (200 mg/kg i.p.; Keppra, UCB Laboratories, Brussels, Belgium) was administered to the rats. The pumps released LEV continuously (~200 mg/kg/day) for 1 week.

### 2.5. Measurement of Levetiracetam Blood Levels

On the last day of treatment, blood samples were obtained via puncture of the caudal vein of the rat in order to measure the LEV concentration. A quantity of 40–50 μL of blood per spot was deposited on a four-well Guthrie card (Whatman 903, Maidstone, UK). LEV was extracted from the Guthrie card and analyzed using high-performance liquid chromatography (HPLC; Alliance HPLC bioseparation system module 2796, Waters, Millford, MA, USA) according to the method published by Oláh et al. [32,36].

### 2.6. Preparation of Biological Samples

The brains were removed and sectioned to obtain the hippocampus, and the tissues were frozen and stored at −80 °C until use. Both structures of hippocampus were homogenized in phosphate buffer 50 mM with 0.05% Triton X-100 (Sigma, St. Louis, MO, USA) at pH = 7.0 in proportion 1:5 using a homogenizer (OMNI International TH, TH-01, Kennesaw, GA USA) and then centrifuged at 15,000× *g* for 30 min at 4 °C. The supernatants were separated and stored in conical tubes for use in the determination of the antioxidant and oxidant markers. It is important to mention that on sacrifice day, none of the animals of the EPI or EPI + LEV groups presented SRS.

### 2.7. Antioxidant Markers

#### 2.7.1. SOD Activity Assay

The activity of SOD and the calculation of its determination were carried out as described by [21] using a superoxide dismutase activity kit (Enzo Life Sciences^®^, Butler Pike Plymouth Meeting, PA, USA). In brief, 25 μL of hippocampal supernatant diluted 1:4, blank or SOD standard; 150 μL of Master Mix (10X SOD buffer, WST-1 reagent, xanthine oxidase, and distilled water); and 25 μL of 1X xanthine solution were added to each well to initiate the reaction. The absorbance was measured at 450 nm every minute for 10 min at room temperature. The data were expressed in U/mg of protein [21].

#### 2.7.2. CAT Activity Assay

The activity of CAT and the calculation of its determination were carried out as described by [21] using a catalase activity kit (Enzo Life Sciences^®^, Butler Pike Plymouth Meeting, PA, USA). In brief, 50 μL of hippocampal supernatant diluted 1:15, blank or CAT standard, and 50 μL of 40 μM of H_2_O_2_ solution were added to each well, and the plate was incubated for 30 min at room temperature. Then, 100 μL of the reaction cocktail (detection reagent, 100X HRP, and 1X reaction buffer) was added to the well and the plate was incubated for 10 min. Fluorescence was measured using excitation and emission wavelengths of 530 and 590 nm, respectively. The data were expressed in U/mg of protein [21].

#### 2.7.3. GPx Activity Assay

The activity of GPx and the calculation of its determination were carried out as described by [21] using a glutathione peroxidase activity kit (Enzo Life Sciences^®^, Butler Pike Plymouth Meeting, PA, USA). In brief, 20 μL of hippocampal supernatant diluted 1:4, blank or GPx standard; 20 μL of 10X reaction mix; 140 μL of 1X assay buffer; and 20 μL of cumene hydroperoxide were added to each well to initiate the reaction. The absorbance was measured at 340 nm every minute for 10 min at room temperature. The data were expressed in U/mg of protein [21].

#### 2.7.4. GR Activity Assay

The activity of GPx and the calculation of its determination were carried out as described by [21] using a glutathione reductase activity kit (Enzo Life Sciences^®^, Butler Pike Plymouth Meeting, PA, USA). In brief, 50 μL of hippocampal supernatant diluted 1:5, blank or GR standard; 100 μL of Master Mix (10X GR buffer, GSSG reagent, and distilled water); and 100 μL of NADP solution were added to each well to initiate the reaction. The absorbance was measured at 340 nm every minute for 10 min at room temperature. The data were expressed in U/mg of protein [21].

### 2.8. Oxidant Markers

#### 2.8.1. H_2_O_2_ Level Determination

H_2_O_2_ levels were determined as described by [21] using a hydrogen peroxide colorimetric detection kit (Enzo Life Sciences^®^, Butler Pike Plymouth Meeting, PA, USA). In brief, 50 μL of hippocampal supernatant diluted 1:8, blank or H_2_O_2_ standard; 50 μL of sample diluent; and 100 μL of color reagent were added per well, and the plate was incubated for 30 min at room temperature. The absorbance was measured at 550 nm. H_2_O_2_ levels were expressed in ng/mg of protein [21].

#### 2.8.2. Protein Carbonylation Determination 

Carbonylated protein content in the samples was determined using the method set out by [37]. Briefly, the homogenates were incubated with 10% streptomycin sulfate for 30 min to remove nucleic acids. In addition, the homogenates were treated with 10 mM of 2,4-dinitrophenyldidrazine (DNPH) (Sigma, St. Louis, MO, USA) and 2.5 M HCl (JT Baker, Xalostoc, Edo. Mexico, Mexico) and, after 3 washes with (1:1) ethanol/ethyl acetate mixture (JT Baker, Xalostoc, Edo. Mexico, Mexico), the protein carbonyl pellet was resuspended in 6 M guanidine hydrochloride (Sigma, St. Louis, MO, USA). Assessment of carbonyl formation was performed on the basis of the formation of protein hydrazone by reaction with DNPH. The absorbance was measured at 370 nm. Protein carbonyl content was expressed as nmol of carbonylated proteins/mg protein [37,38].

#### 2.8.3. MDA Level Determination

The MDA content was measured using a standard curve of tetramethoxypropane. Supernatants (70 μL) (Sigma, St. Louis, MO, USA) were added to 0.25 mL of 10 mM 1-methyl-2-phenylindole (Sigma, St. Louis, MO, USA) in a mixture of acetonitrile/methanol (3:1) (JT Baker, Xalostoc, Edo. Mexico, Mexico). The reaction was started by adding 50 μL of 37% HCl (JT Baker, Xalostoc, Edo. Mexico, Mexico) and incubating for 40 min at 45 °C. Then, the samples were centrifuged at 3000× *g* for 5 min. The optical density of the supernatant was measured at 586 nm. Data were expressed as nmol MDA/mg of protein [39,40].

#### 2.8.4. Total Protein Measurement

The protein content in the samples was determined using the Lowry method. Briefly, the protein quantities in these samples were assessed using an 8-point standard curve of bovine serum albumin (Sigma, St. Louis, MO, USA), which was used as a reference standard. Samples were diluted and transferred to the microplate, 250 μL of the following solution: 2% Na_2_CO_3_ (JT Baker, Xalostoc, Edo. Mexico, Mexico), 0.4% NaOH (JT Baker, Xalostoc, Edo. Mexico, Mexico), 0.02% C_4_H_4_O_6_KNa·4H_2_O (Mallinckrodt, Staines, UK), and 0.01% CuSO_4_ (JT Baker, Xalostoc, Edo. Mexico, Mexico) was added to each well. The mixture was then allowed to incubate at room temperature for 10 min prior to the addition of 25 μL per well of 1.0 N Folin and Ciocalteu’s reagent (Sigma, St. Louis, MO, USA). Samples were mixed immediately and after 30 min, absorbance was measured at 660 nm [41]. 

### 2.9. In Vitro Scavenging Activity of LEV

#### 2.9.1. DPPH Assay

DPPH radical scavenging activity was investigated according to the method presented by [42]. Briefly, 70 µL of dilutions of LEV (Sigma, St. Louis, MO, USA) were mixed with 50 µL of 2 mM DPPH (Sigma, St. Louis, MO, USA). After 2 min, 800 µL of ethanol (JT Baker, Xalostoc, Edo. Mexico, Mexico) was added, and the mixture was kept for 2 min at room temperature. Optical density at 517 nm was recorded using a Synergy HT1 multimode microplate reader (Biotek Instruments Inc. Winooski, VT, USA) [42]. DPPH scavenging percentage was determined from Equation (1), considering the absorbance of control (A0) and the sample (AX).
% DPPH scavenging = ((A0 − AX)/AX) × 100)(1)

#### 2.9.2. FRAP Measurement

The antioxidant capacity of a molecule is revealed when it reacts with the colorless ferric tripyridyltriazine (Fe^3+^-TPTZ) complex and forms blue ferrous tripyridyltriazine (Fe^2+^-TPTZ). Briefly, 30 µL of LEV (Sigma, St. Louis, MO, USA) was mixed with 300 µL of FRAP reagent [0.833 mM 2,4,6-Tris(2-pyridyl)-s-triazine (TPTZ) (Sigma, St. Louis, MO, USA), 1.66 mM FeC_l3_ (Sigma, St. Louis, MO, USA), in 250 mM acetate buffer, pH 3.6] and after 15 min, the absorbance at 593 nm was measured using a Synergy HT multi-mode microplate reader (Biotek Instruments Inc. Winooski, VT, USA). Results were expressed as Trolox equivalent (Sigma, St. Louis, MO, USA)/g of the sample [43].

#### 2.9.3. ORAC Assay

This antioxidant determination was made according to the method described by [44]. Briefly, samples comprising 25 µL of water, Trolox (Sigma, St. Louis, MO, USA) standards, and diluted LEV (Sigma, St. Louis, MO, USA) were mixed with 25 µL of 153 mM (2,2-azobis(2-amidinopropane) dihydrochloride) (AAPH) (Sigma, St. Louis, MO, USA) and 150 µL of 50 nM fluorescein (Sigma, St. Louis, MO, USA) and incubated at 37 °C. The fluorescence was recorded every minute for 90 min using fluorescence filters for excitation and emission wavelengths of 485 nm and 520 nm, respectively, using a Synergy HT1 multimode microplate reader (Biotek Instruments Inc. Winooski, VT, USA). The ORAC values were calculated using the net area under the decay curves and were expressed as micromoles of Trolox equivalents/g of the sample [44].

#### 2.9.4. HO• Scavenging Assay

The HO• was generated by the Fe^3+^–EDTA–H_2_O_2_ reaction, and terephthalic acid (TA) was used to assess the generation of radicals because the nonfluorescent compound TA reacts with HO• to form fluorescent 2-hydroxy-TA. In brief, 180 µL of a reaction mixture comprising 0.2 mM ascorbic acid ((Sigma, St. Louis, MO, USA), 0.2 mM FeCl_3_ (Sigma, St. Louis, MO, USA), 0.208 mM EDTA (Sigma, St. Louis, MO, USA), 1 mM H_2_O_2_ (JT Baker, Xalostoc, Edo. Mexico, Mexico), and 1.4 mM terephthalic acid (Sigma, St. Louis, MO, USA), in 20 mM phosphate buffer (pH 7.4), was mixed with 20 µL of different concentrations of LEV (Sigma, St. Louis, MO, USA). The increase in the fluorescence signal was measured for 30 min at excitation and emission wavelengths of 326 nm and 432 nm, respectively [45]. The scavenging percentage was determined from Equation (2), considering the fluorescence of control (F0) and the sample (FX).
% HO• scavenging = ((F0 − FX)/FX) × 100)(2)

#### 2.9.5. O_2_^•−^ Scavenging Capacity

The xanthine–xanthine oxidase (XO) system was used to determine the O_2_^•−^ scavenging capacity of the samples based on the method described by [46]. O_2_^•−^ in the assay system and XO activity were measured as nitroblue tetrazolium salt (NBT) reduction and uric acid production, respectively. This system is a useful test for O_2_^•−^ scavenging capacity only when the samples used do not interfere with the XO activity. A compound with O_2_^•−^ scavenging capacity should decrease NBT reduction without interfering with XO activity which is measured as uric acid production. Briefly, 180 µL of a reaction mixture comprising 90 µM xanthine (Sigma, St. Louis, MO, USA), 16 mM Na_2_CO_3_ ((JT Baker, Xalostoc, Edo. Mexico, Mexico), 22.8 µM NBT (Sigma, St. Louis, MO, USA), and 18 mM phosphate buffer (pH 7.0) was mixed with 20 µL of different concentrations of LEV (Sigma, St. Louis, MO, USA. The reaction was started by the addition of 10 µL of XO (0.1 U/mL)(Sigma, St. Louis, MO, USA). Optical density was registered both at 295 nm (for uric acid production) and 560 nm (for O_2_^•−^ in the assay system) using a Synergy HT1 multimode microplate reader (Biotek Instruments Inc. Winooski, VT, USA). The scavenging percentage was obtained from the optical densities at 560 nm [46]. Scavenging percentage was obtained from the optical densities at 560 nm [46]. The scavenging percentage was determined from Equation (3), considering the absorbance of control (A0) and the sample (AX).
%O_2_^•−^ scavenging = ((A0 − AX)/AX)) × 100)(3)

### 2.10. Statistical Analysis

For the animal experiments, all data are presented as the mean ± standard deviation (SD) for the animals in each group (*n* = 6). The Shapiro–Wilk normality test was performed based on the null hypothesis that the distribution is normal. To determine differences in antioxidant enzymes and oxidative stress markers between the groups, data were analyzed by one-way ANOVA and Tukey’s post hoc test. The determinations were performed in triplicate. A *p*-value < 0.05 was considered to indicate a significant difference. All data were analyzed using Sigma Plot (USA). The in vitro HO• scavenger capacity is presented as mean ± SD (*n* = 3).

## 3. Results

### 3.1. Levetiracetam Blood Levels

Seven days after micropump implantation, the LEV levels in the blood were 26.17 ± 2.07 and 27.82 ± 3.53 μg/mL (mean ± SD) in the CTRL + LEV and EPI + LEV groups, respectively, without significant differences between these groups. No significant concentrations of LEV were detected in the CTRL and EPI groups. These data are consistent with the findings of previous studies [32].

### 3.2. Antioxidant Markers

The SOD activity levels measured for the CTRL, CTRL + LEV, EPI, and EPI + LEV groups were 63.97 ± 2.77, 64.42 ± 3.27, 54.42 ± 2.98, and 79.18 ± 4.91 units/mg of protein, respectively. SOD activity was significantly higher (1.45-fold) in the EPI + LEV group in comparison with the EPI group (*p* < 0.01). No significant changes were observed for the CTRL and CTRL + LEV groups (Figure 1A). The CAT activity measured for the CTRL, CTRL + LEV, EPI, and EPI + LEV groups were 9.636 ± 2.3, 19.86 ± 4.9, 4.46 ± 0.68, and 12.81 ± 3.69 units/mg of protein, respectively. CAT activity in the CTRL + LEV group was significantly higher (4.45-fold) in comparison with the EPI group (*p* < 0.05) (Figure 1B). GPx activity levels measured for the CTRL, CTRL + LEV, EPI, and EPI + LEV groups were 73.26 ± 3.08, 66.65 ± 2.50, 87.84 ± 6.43, and 91.52 ± 6.27 units/mg of protein, respectively. GPx activity in the EPI + LEV group was significantly higher (1.37-fold) in comparison with the CTRL + LEV group (*p <* 0.05; Figure 1C). GR concentrations measured for the CTRL, CTRL + LEV, EPI, and EPI + LEV groups were 48.63 ± 5.29, 49.39 ± 4.07, 127.63 ± 7.27, and 48.42 ± 4.75 munits/mg of protein, respectively. GR activity in the EPI group was significantly higher (2.6-fold) in comparison with all the other groups (*p <* 0.01) (Figure 1D).

In the comparative analysis, the following observations were made for the EPI group: a significant decrease in SOD activity and an increase in GR activity in comparison with the EPI + LEV group; a decrease in CAT activity and an increase in GR activity compared with the CTRL + LEV group; and an increase in GR activity compared with the CTRL group. In the EPI + LEV group, GPx activity was increased in comparison with the CTRL + LEV group. Additionally, it was observed that in the EPI + LEV group, CAT and GR activity were similar to their respective CTRL groups, and in the CTRL + LEV group, SOD, GPx, and GR activities were similar to their respective CTRL group. In the EPI group it was also observed that CAT activity showed a nonsignificant decrease in comparison with the EPI + LEV group.

### 3.3. Oxidant Markers

For H_2_O_2_ concentration, the CTRL, CTRL + LEV, EPI, and EPI + LEV groups had 178.55 ± 10.27, 343.08 ± 29.28, 409.69 ± 9.56, and 190.75 ± 17.09 ng/mg of protein, respectively. In the CTRL + LEV group, H_2_O_2_ levels were significantly higher (1.9-fold) in comparison with the CTRL group and significantly higher (1.8-fold) in comparison with the EPI + LEV group (*p* < 0.01 for both). Similarly, in the EPI group, the H_2_O_2_ levels were significantly higher (2.3-fold) in comparison with the CTRL group and significantly (2.14-fold) higher compared with the EPI + LEV group (*p* < 0.01 for both; Figure 2A). For carbonylated protein concentration, the CTRL, CTRL + LEV, EPI, and EPI + LEV groups had 12.90 ± 0.89, 12.15 ± 0.37, 13.46 ± 0.95, and 10.06 ± 0.83 nmol/mg of protein, respectively. For MDA concentration, the CTRL, CTRL + LEV, EPI, and EPI + LEV groups had 6.75 ± 0.17, 6.75 ± 0.08, 6.19 ± 0.46, and 5.72 ± 0.71 nmol/mg of protein, respectively (Figure 2C). Nonsignificant differences in carbonylated protein and MDA concentrations were observed (Figure 2B,C).

In the comparative analysis, a significant increase in H_2_O_2_ levels was observed in the EPI group in comparison with the CTRL and EPI + LEV groups. A significant increase in H_2_O_2_ levels was observed in the CTRL + LEV group in comparison with the CTRL and EPI + LEV groups. No significant changes were observed in lipoperoxidation and carbonylated protein levels.

### 3.4. In Vitro Scavenging Activity of LEV

This was the first time that the in vitro scavenging capacity of LEV for DPPH, HO•, and O_2_^•−^, FRAP, and ORAC had been determined. For the scavenging capacity we calculated the IC_50_ value, which denotes the concentration of LEV required to give a 50% reduction in the oxidizing effect relative to the blank tube. LEV did not display scavenging capacity against DPPH and O_2_^•−^ but it showed antioxidant activity against HO•, obtaining an IC_50_ (mM) of 2751 ± 0.086. In addition, LEV showed no FRAP or ORAC potential.

## 4. Discussion

In this study, the results revealed that epilepsy causes an imbalance between the pro- and antioxidant systems and that the administration of LEV partially restores this imbalance. Changes in OS markers have been observed in many human and experimental studies; for example, there have been reports of increases in lipid peroxidation (MDA or thiobarbituric acid reactive substances (TBARS) levels), protein nitration (peroxynitrite, ONOO^−^ 3-nitrotyrosine, 3-NT or nitric oxide, NO• levels) or carbonylation (carbonylated protein levels), DNA damage (8-hydroxy-2-deoxyguanosine), or in markers of total antioxidant status or OS index. Accompanying these changes, decreases in the levels of antioxidant markers, such as reduced glutathione (GSH), CAT (reduction of H_2_O_2_ to H_2_O), GPx (reduction of peroxides to H_2_O), SOD (reduction of O_2_^•−^ to H_2_O_2_), or GR (reduction of oxidized GSSG to GSH) have been observed [21,47,48,49].

In the development of TLE, the involvement of OS and neuroinflammation have been well established as important factors, in part because the large amount of energy required by the brain is considered to be a possible mechanism involved in epileptogenesis [12,21,50,51,52,53,54,55,56,57]. In recent studies, it was observed that the specific inhibition of NADPH oxidase (NOX), the main producer of OS through O_2_^•−^ synthesis, modified chronic epilepsy in a TLE rat model, preventing ROS generation, mitochondrial depolarization, and neuronal death. The finding suggests that this enzyme contributes to epilepsy development [58] and that overexpression of the cAMP response element binding protein (CREB) protects against mitochondrial OS, apoptosis, and cognitive dysfunction in a TLE mouse model [59].

In our results, we observed no significant changes in lipoperoxidation and carbonylated protein levels in EPI rats in comparison with the CTRL group. However, H_2_O_2_ levels were significantly increased in EPI rats in comparison with the CTRL and EPI + LEV groups. The increase in H_2_O_2_ levels in TLE observed in our results is consistent with the literature. In a TLE model, it was demonstrated that H_2_O_2_ was released during SE and correlated with changes in electrical activity. The increase in H_2_O_2_ levels was observed after 15 days of SE, and this increase persisted during the chronic period, indicating that this reactive species could participate in the generation and maintenance of seizures [60]. Another study detected an increase in 8-OHdG in mitochondria isolated from the hippocampus of rats with TLE immediately after SE. Additionally, an increase in H_2_O_2_ levels after SE and epileptogenesis was observed [61].

With regard to antioxidant enzyme activity, we observed a decrease in CAT, a slight decrease in SOD, a significant increase in GR, and a slight increment in GPx in the epileptic rats compared with the CTRL group. In a recent study, an increase in the expression of the nuclear factor erythroid 2-related (Nrf2) was observed in the hippocampus during epileptogenesis in a rat model of TLE. Nrf2 is known as the master inductor of the expression of antioxidant and detoxification enzymes and genes involved in mitochondrial biogenesis and preservation. The authors concluded that the activation of Nrf2 mediated the antioxidant response after brain insult and that it could therefore modify the development of epilepsy [62,63] (Figure 3). Based on the above, it is probable that the nonsignificant changes in the increment of lipoperoxidation and protein oxidation observed in epileptic rats were due to Nrf2 activation, which controls the redox changes in the hippocampus during TLE, and also to the modulating activities of GR and GPx, which prevent OS from reaching the oxidation of the protein and lipids. This would also explain the significant increase in GR activity in the epileptic brain, as this enzyme is the main source of GSH synthesis. GSH is recognized as the most abundant antioxidant and plays a crucial role in the maintenance of redox homeostasis in neurons [64]. This finding is supported by a study of the hippocampi of patients with pharmacoresistant TLE, using autopsies of dead subjects as controls, in which an increase in GR and GPx activities and protein levels were observed in the TLE patients compared with the control subjects. The authors of this clinical study stated that subjects with the TLE condition were exposed to excessive OS, mediated via mitochondrial O_2_^•−^ and H_2_O_2_ because of a dysfunction in mitochondrial complex I in CA neurons. They also mentioned that this dysfunction affects CAT and GPx profiles because CAT and GPx/GR represent the main sinks for H_2_O_2_ in mitochondria [65,66,67,68], being necessary the increment in GR and GPx activities in a brain with seizures, as we also observed. We should mention the importance of the study and limitations of brain imaging or electroencephalographic methods for the distinction between the different types of epilepsy and for the research of epilepsy in human and animal models due to the need to perform correlation of different biochemical observations or diagnoses between the human brain or rat brain, as was performed in the present study [69,70].

The key findings in our work reveal the antioxidant properties of LEV and that administration of the drug in epileptic rats significantly decreased H_2_O_2_ and significantly increased SOD activities in comparison with the EPI group. There was also no significant increase in CAT activity in comparison with the EPI group. In the CTRL + LEV group, CAT activity showed no significant increase in comparison with the CTRL group. LEV is a known ASD, and its principal molecular mechanism is SV2A modulation. As well as being an anticonvulsant agent, several studies have shown its other potential uses as an antihyperalgesic and anti-inflammatory drug for the treatment of motor diseases, cognitive dysfunctions, and mental disorders, in addition to its use as an anticancer drug, and, recently, as an antioxidant [26]. In a few studies it has been shown that LEV possesses antioxidant properties, increasing antioxidant markers, such as GSH, SOD, and GPx activities, and decreasing oxidant markers, such as TBARS, nitrite/nitrate, 8-OHdG, and 8-isoprostanes levels, in experimental models of epilepsy [20,27,28,29,71,72]. Our work is the first to demonstrate the antioxidant properties of LEV in a TLE experimental model.

The exact mechanism via which LEV induces the antioxidant effect remains unclear; however, there are some hypotheses. It has been shown that LEV induces the phosphorylation of the phosphoinositide-3-kinase (PI3K) and serine/threonine protein kinase (Akt) pathways, which are related to the amelioration of OS and DNA damage, inflammation, autophagy, hippocampal neurogenesis, and increased neuronal survival in TLE [73,74,75,76]. LEV also activates Nrf2 signaling, allowing an increase in the upregulation by GSH of the cystine/glutamate exchange transporter in the hippocampus of rats [27,77]. In the hippocampus, NMDA receptors promote a calcium inflow into synapses, which activates protein kinases such as PI3K and protein kinase C (PKC), leading to ROS production as NO• through neural nitric oxide synthase (nNOS) and H_2_O_2_. The H_2_O_2_ can pass through the plasmatic membrane and modulate synaptic transmission but also produce greater neuronal damage in a pathological state such as TLE [78,79]. Another experimental work with a TLE rat model showed the presence of hyperexcitability in the hippocampus of rats, and this condition was associated with tumor necrosis factor-α (TNF-α). The authors suggested that TNF-α could be producing O_2_^•−^ and H_2_O_2_, which caused OS [80], and that in the hippocampus, the PI3K/AKT/mTOR pathway was inhibited, SOD activity decreased, and apoptosis and lipid oxidation increased [75]. These observations are consistent with another study in which LEV in the hippocampus of nonepileptic (normal) mice improved cell proliferation and expression of SOD and CAT and increased PI3K/Akt in comparison with the control group [81]. LEV could be exerting an indirect antioxidant role that decreases H_2_O_2_ and O_2_^•−^ levels through the modulation of PI3K activity and induction of CAT (reducing H_2_O_2_ levels) and SOD (reducing O_2_^•−^ to H_2_O_2_) activities via the Nrf2 pathway in the hippocampus with TLE, and which also causes a reduction in the oxidant status (Figure 3). Another possible antioxidant mechanism is the activation of CREB expression through LEV (in TLE, the expression of this transcription factor is decreased). This was shown in a chronic cerebral hypoperfusion model in which LEV significantly increased the expression of SV2A and pCREB, and OS was reduced in the brain white matter [82]. These observations add weight to the idea that LEV has a role in positively modulating SOD and CAT activities during TLE.

On the other hand, we observed significantly decreased GR activity in the EPI + LEV rats in comparison with the EPI rats, and significantly increased GPx activity in comparison with the CTRL + LEV group. However, there was no change in GPx activity in comparison with the EPI group. Changes in the mitochondrial membrane can potentially cause subsequent changes in neuronal excitability. However, considering that seizures induced an alteration in antioxidant defenses, LEV probably has no effect to induce a positive modulation or an increase of GR and GPx activities in the dysfunctional hippocampus of TLE rats [65,83]. However, it was observed that EPI + LEV rats displayed significantly increased GPx activity in comparison with the CTRL + LEV group, indicating that this enzyme is probably activated to detoxify the increment of H_2_O_2_ levels in the epileptic condition. Furthermore, in the case of the activity of GR, the EPI + LEV group showed similar values to the CTRL and CTRL + LEV groups, which suggested that treatment with LEV could be reestablish the antioxidant system to basal levels through GR, being beneficial because the imbalance in the redox state can result in altered neuronal homeostasis (Figure 3). 

Finally, our results revealed, for the first time, that LEV shows scavenging activity against HO•. From the hydroxylation of the pyrrolidone ring in the metabolism of LEV it has been identified that LEV probably interacts with HO• (one of the principal products derived from H_2_O_2_ through the univalent reduction of oxygen), and, through this functional group, acts as a reductor agent for this radical, preventing the formation of other ROS due to HO• being recognized as the most reactive radical and the major source of cellular damage [84,85]. This result could indicate that LEV displays direct antioxidant activity and that this characteristic participates in the modulation of the redox state in the TLE model.

## 5. Conclusions

LEV showed antioxidant effects, significantly increasing SOD activity, increasing CAT activity, and decreasing H_2_O_2_ levels in the hippocampus of rats, with TLE restoring the redox balance. Additionally, LEV showed in vitro scavenging against HO•, indicating a direct antioxidant activity. However, more studies in experimental models and human clinical trials are required to demonstrate the antioxidant capacity of LEV and also to explain the biochemical and molecular mechanisms underlying the direct and indirect antioxidant effects of this drug.

## Figures and Tables

**Figure 1 biomedicines-11-00848-f001:**
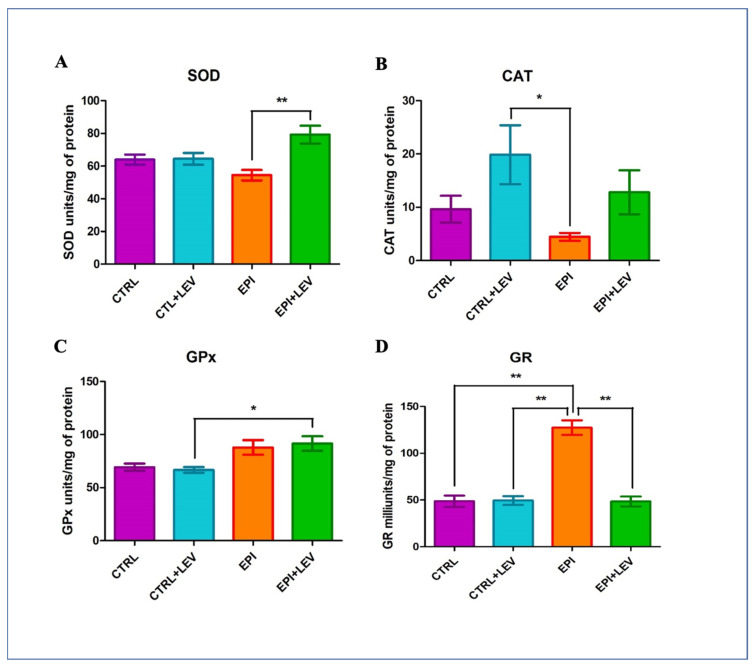
Antioxidant markers determined in the hippocampus of the rat. In (**A**–**D**) are showed the antioxidant activities of SOD, CAT, GPx and GR, respectively. For all measurements, each quantification was performed in triplicate, using data from six rats for each group; values plotted represent mean ± SD. Differences were analyzed using one-way ANOVA and Tukey’s post hoc test. The brackets indicate the groups which are significantly different. * *p* < 0.05 represents significant differences. ** *p* < 0.01 represents significant differences.

**Figure 2 biomedicines-11-00848-f002:**
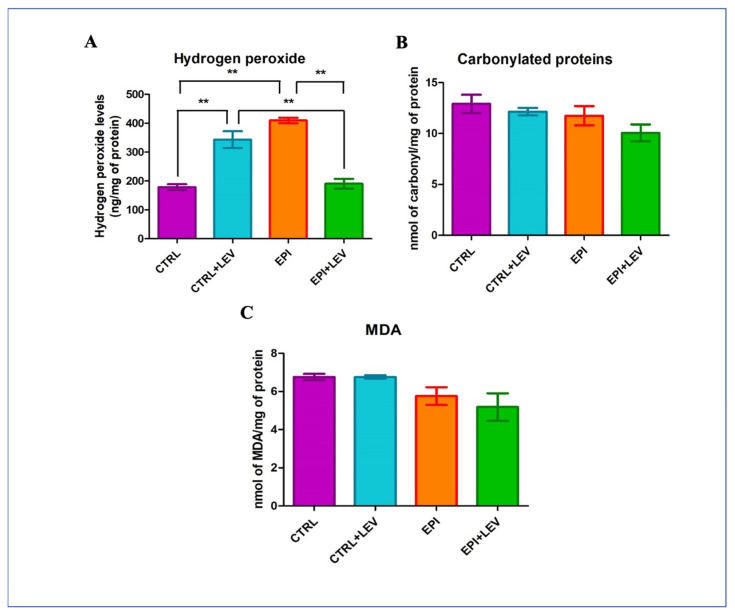
Oxidant markers determined in the hippocampus of the rat. In A, B and C are showed the concentrations of H_2_O_2_, carbonylated proteins and MDA, respectively For all measurements, each quantification was performed in triplicate, using data from six rats for each group; values plotted represent mean ± SD. Differences were analyzed using one-way ANOVA and Tukey’s post hoc test. The brackets indicate the groups that are significantly different. ** *p* < 0.01 represents significant differences.

**Figure 3 biomedicines-11-00848-f003:**
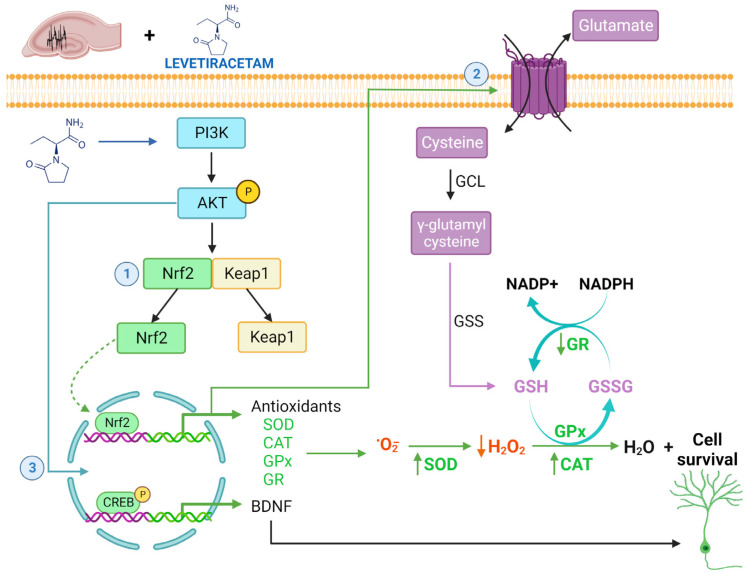
Hypothetical antioxidant action of levetiracetam (LEV) in the epileptic brain. Administration of LEV in the epileptic brain may induce (1) the phosphorylation of phosphoinositide-3-kinase (PI3K) and serine/threonine protein kinase (Akt), allowing the nuclear factor erythroid 2-related factor 2 (Nrf2) translocation to nuclei, which increases SOD and CAT enzyme activities, resulting in a reduction in O_2_^•−^ (by SOD) and H_2_O_2_ (by CAT) levels; (2) upregulation of the cystine/glutamate exchange transporter by Nrf2 signaling, increasing the GSH enzyme (involved in the antioxidant pathway); and (3) activation of the transcription factor CREB which is involved in brain-derived neurotrophic factor (BDNF) expression and, subsequently, cell survival.

## Data Availability

The data presented in this study are available on request from the corresponding author.

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
