# Peer review of "Evaluation of the Antioxidant Activity of Levetiracetam in a Temporal Lobe Epilepsy Model"

_biomedicines, 2023, doi:10.3390/biomedicines11030848_

Round 1
Reviewer 1 Report
There are a few matters that I would like to have seen considered in the paper as they might cause a degree of reconsideration in a few of the statements in the Discussion section, but they may have merely been overlooked.
Results are often expressed on a denominator of protein content, but I do not see how the latter was measured and assume it was measured in the same specimen as the substance investigated.
The paper speaks of the hippocampus, but this is a bilateral structure. Were the measurements all made on the hippocampus on the same side, on both sides, or on either side, at random? This could be quite relevant if you are going to claim that your experimental preparation is a model of temporal lobe epilepsy. This, in humans, as a focal epilepsy often with the structural correlate of hippocampal neuronal loss and gliosis on the epileptogenic side. I saw no evidence in the paper that you had determined and then studied the side in which epileptogenesis began in your animals, and I suspect that with your chemical induction method the seizure production may have had bilateral effects.
Were you animals still having spontaneous seizures at or close to the time of their demise? If so, is it possible that your measurements reflect consequences of seizure activity rather than of the underlying epileptogenic process? The answer to this might alter your interpretation of the results.
In your abstract you write of epilepsy as a `pathology’? I wonder if the word ‘disorder’ would be safer since some forms of epilepsy are not associated with detectable pathology. Also, there seems to be an increasing tendency these days to write of ‘antiseizure medication’ rather than ‘antiepileptic drugs’.
Author Response
Comment
There are a few matters that I would like to have seen considered in the paper as they might cause a degree of reconsideration in a few of the statements in the Discussion section, but they may have merely been overlooked.
Response
Thank you for your valuable comments to improve the manuscript
Results are often expressed on a denominator of protein content, but I do not see how the latter was measured and assume it was measured in the same specimen as the substance investigated.
Response
The protein content was measured in hippocampus (study specimen) we added the protein content method in the manuscript. Thank you by your observation.
Comment
The paper speaks of the hippocampus, but this is a bilateral structure. Were the measurements all made on the hippocampus on the same side, on both sides, or on either side, at random? This could be quite relevant if you are going to claim that your experimental preparation is a model of temporal lobe epilepsy. This, in humans, as a focal epilepsy often with the structural correlate of hippocampal neuronal loss and gliosis on the epileptogenic side. I saw no evidence in the paper that you had determined and then studied the side in which epileptogenesis began in your animals, and I suspect that with your chemical induction method the seizure production may have had bilateral effects.
Response
The determinations were realized in both structures of hippocampus and we did not determine the side of the hippocampus where epileptogenesis was initiated in the rat due to our experimental design, it is not possible to say for sure that the epileptogenesis process begins in one of the temporal lobes . In relation to the chemical induction of TLE we used the lithium-pilocarpine that is a method that has been showed the induction of acute SE (for several hours), the presence of a latent period followed by the appearance of spontaneous recurrent seizures (chronic phase follows epileptogenesis) and the occurrence of widespread lesions some of them localized in the same brain areas affected in TLE patients, and associated with neuronal network reorganization in hippocampal and parahippocampal regions in both regions as you say. In the manuscript we added that the determinations were realized in both structures of hippocampus .
Comment
Were you animals still having spontaneous seizures at or close to the time of their demise? If so, is it possible that your measurements reflect consequences of seizure activity rather than of the underlying epileptogenic process? The answer to this might alter your interpretation of the results.
Response
Thank you for your comment. On day of sacrifice, none of the animals presented recurrent spontaneous seizures, since, as mentioned, this could modify the levels of the measured variables. This information was included in the sections Materials and Methods at lines 224 and 225 (page 4).
Comment
In your abstract you write of epilepsy as a `pathology’? I wonder if the word ‘disorder’ would be safer since some forms of epilepsy are not associated with detectable pathology. Also, there seems to be an increasing tendency these days to write of ‘antiseizure medication’ rather than ‘antiepileptic drugs’.
Response
The terms were changed in the manuscript: “pathology” was changed as “disorder” in the abstract and “antiepileptic drugs (AED)” was changed as “antiseizure drugs (ASD)” in accord with the recommendations of International League Against Epilepsy (ILAE) of 2022 throughout the manuscript. Thank you for this observation
Reviewer 2 Report
At the manuscript "Evaluation of the antioxidant activity of levetiracetam in a temporal lobe epilepsy model” by Drs. Iván Ignacio-Mejía et al authors describe results of evaluation of the effects of
Levetiracetam (LEV) on the oxidant–antioxidant status in the hippocampus of rats with temporal lobe epilepsy (TLE). It was demonstrated that LEV administration in rats with TLE significantly increased superoxide dismutase (SOD) activity, increased catalase (CAT) activity but did not change glutathione peroxidase (GPx) activity and significantly decreased glutathione reductase (GR) activity in comparison with epileptic rats.
Also, it was shown that LEV administration in rats with TLE significantly reduced hydrogen peroxide (H2O2) levels but did not change lipoperoxidation and carbonylated protein levels in comparison with epileptic rats. In addition, LEV showed in vitro scavenging activity against hydroxyl radical (HO). LEV showed significant antioxidant effects in relation to restoring the redox balance in the hippocampus of rats with TLE.
As regards in vitro studies, LEV demonstrated direct antioxidant activity against HO.
The authors have done a great and serious study, and I have no objections in essence.
I have only minor criticisms:
The authors write that OS, ROS, or RNS and inflammation promote neuronal hyperexcitability and seizures and are crucial factors in the onset and progression of neurodegeneration in temporal lobe epilepsy (TLE), which is one of the most common epilepsies of focal origin [8-12]. This is absolutely true, but raises the question of absence epilepsy? Could the authors add a few sentences about this?
The authors quite rightly cite papers on imaging, but there are publications on imaging of absence epilepsy, they should be included in the discussion:
Lenkov et al, Advantages and limitations of brain imaging methods in the research of absence epilepsy in humans and animal models J Neurosci Methods . 2013 30;212(2):195-202.
van Luijtelaar et al, Is there such a thing as "generalized" epilepsy? Adv Exp Med Biol. 2014;813:81-91. doi: 10.1007/978-94-017-8914-1_7.
The presentation of a subject is systematic and comprehensive and conclusion is proper. I am happy to recommend the manuscript for the publication after minor corrections mentioned above.
Author Response
Comment
At the manuscript "Evaluation of the antioxidant activity of levetiracetam in a temporal lobe epilepsy model” by Drs. Iván Ignacio-Mejía et al authors describe results of evaluation of the effects of Levetiracetam (LEV) on the oxidant–antioxidant status in the hippocampus of rats with temporal lobe epilepsy (TLE). It was demonstrated that LEV administration in rats with TLE significantly increased superoxide dismutase (SOD) activity, increased catalase (CAT) activity but did not change glutathione peroxidase (GPx) activity and significantly decreased glutathione reductase (GR) activity in comparison with epileptic rats.
Also, it was shown that LEV administration in rats with TLE significantly reduced hydrogen peroxide (H2O2) levels but did not change lipoperoxidation and carbonylated protein levels in comparison with epileptic rats. In addition, LEV showed in vitro scavenging activity against hydroxyl radical (HO). LEV showed significant antioxidant effects in relation to restoring the redox balance in the hippocampus of rats with TLE.
As regards in vitro studies, LEV demonstrated direct antioxidant activity against HO.
The authors have done a great and serious study, and I have no objections in essence.
Response
Thank you for your valuable comment
Comment
I have only minor criticisms:
The authors write that OS, ROS, or RNS and inflammation promote neuronal hyperexcitability and seizures and are crucial factors in the onset and progression of neurodegeneration in temporal lobe epilepsy (TLE), which is one of the most common epilepsies of focal origin [8-12]. This is absolutely true, but raises the question of absence epilepsy? Could the authors add a few sentences about this?
Response
Information related to absence epilepsy and the presence of OS was added in the Introduction.
Comment
The authors quite rightly cite papers on imaging, but there are publications on imaging of absence epilepsy, they should be included in the discussion:
Lenkov et al, Advantages and limitations of brain imaging methods in the research of absence epilepsy in humans and animal models J Neurosci Methods . 2013 30;212(2):195-202.
van Luijtelaar et al, Is there such a thing as "generalized" epilepsy? Adv Exp Med Biol. 2014;813:81-91. doi: 10.1007/978-94-017-8914-1_7.
Response
The papers were added in the discussion. Thank you for this observation.
Comment
The presentation of a subject is systematic and comprehensive and conclusion is proper. I am happy to recommend the manuscript for the publication after minor corrections mentioned above.
Response
Thank you for your kind appreciation.